# A New Genus *Neotricholomopsis* Gen. Nov and Description of *Neotricholomopsis globispora* Sp. Nov. (Phyllotopsidaceae, Agaricales) from Northwestern China Based on Phylogeny, Morphology, and Divergence Time

**DOI:** 10.3390/jof10110784

**Published:** 2024-11-13

**Authors:** Longfei Fan, Biyue Wang, Xue Zhong, Hongmin Zhou, Shunyi Yang, Xiaohong Ji

**Affiliations:** 1College of Plant Protection, Gansu Agricultural University, Lanzhou 730000, China; wang_biyue0820@163.com (B.W.); yangshy@gsau.edu.cn (S.Y.); 2Ecology and Nature Conservation Institute, Chinese Academy of Forestry, Beijing 100091, China; zxue021221@163.com; 3College of Biodiversity Conservation, Southwest Forestry University, Kunming 650224, China; hongminzhou@foxmail.com; 4College of Pharmacy and Life Sciences, Jiu Jiang University, Jiujiang 332005, China

**Keywords:** morphology, ITS, nLSU, phylogenetic analysis, divergence time, taxonomy

## Abstract

While investigating macrofungi diversity in Gansu province, northwestern China, five fresh and fleshy specimens were collected, which are characterized by nearly white to buff hemispherical pileus with waved margins, a disc depressed with coral to brownish red fibrillose scales, adnate to sub-decurrent lamellae with four relatively regular rows of lamellulae, a stipe that is central, hollow, frequently straight to curved; basidiospores that are globose to subglobose, 5.0–6.0 × (3.5−) 4.0–5.0 (−5.5) μm, narrowly clavate cheilocystidia predominantly, pleurocystidia and caulocystidia not observed; and a cutis pileipellis, with hyphae slightly inflated in the KOH. The results of phylogeny analysis indicated that the species forms an independent lineage in Phyllotopsidaceae based on the ITS (ITS5/ITS4) and nLSU (LR0R/LR7) dataset. Molecular clock analyses suggested the common ancestor of *Neotricholomopsis* emerged later than upper Cretaceous with a mean crown age of 229.36 Mya (95% highest posterior density of 129.63–343.08 Mya). These five specimens were described as an unreported taxon based on the phylogeny analysis combined with morphological examination and ecological and geographical distribution. Detailed descriptions, illustrations, and phylogenetic trees to demonstrate the placement of this species and discussions with its related species are provided.

## 1. Introduction

The family Phyllotopsidaceae was established by Olariaga et al. to accommodate *Phyllotopsis* E.-J. Gilbert and Donk ex Singer, *Pleurocybella* Singer, and *Macrotyphula* R.H. Petersen in the suborder Pleurotineae, except for *Tricholomopsis* Singer [1]. Until 2023, Wang et al. [2] updated the phylogenetic and taxonomic of Agaricales based on 555 single-copy orthologous genes and delimitated Phyllotopsidaceae with an emphasis on *Tricholomopsis*. Their study accepted *Conoloma* Zhu L. Yang and G.S. Wang, *Phyllotopsis* E.-J. Gilbert and Donk ex Singer, *Pleurocybella* Singer, and *Tricholomopsis* within the family Phyllotopsidaceae of the suborder Phyllotopsidineae. The family is characterized by a fleshy, pleurotoid, or tricholomoid basidioma, cylindrical to clavate basidia with four-spored, smooth and non-amyloid basidiospores, prominent cheilocystidia, and the presence of clamp connections in the basidioma. All species in four genera share a saprotrophic nutrition mode, clamp connections, and a cutis pileipellis with a transition to a trichoderm [2]. It is currently uncertain where *Tricholomopsis* stands in terms of its evolutionary relationships before 2023 [3,4,5,6].

The genus *Tricholomopsis*, typified by *T. rutilans* (Schaeff.) Singer, was created in 1939. The characteristics of the *Tricholomopsis* are a tricholomatoid basidiomata with finely fibrillose scales pileus, adnate to sinuate lamellae, a central stipe without a ring, smooth basidiospores inamyloid, with a cutis to tricholomatoid pileipellis, and the presence of large cheilocystidia and obvious clamp connections [7,8,9].

The distributions of species in *Tricholomopsis* are widespread. Most species that are recorded inhabit conifer woods and usually cause a white rot disease, with partial species growing on bamboo or being terrestrial [9,10,11]. The new taxa *Neotricholomopsis globispora* was only found in coniferous forests. Among them, *T. bambusina* Hongo, *T. decorra* (Fr.) Singer, *T. lividipileata* P.G. Liu, and *T. rutilans* have a vital edible value in China [12,13]. However, it is claimed that *T. rutilans* can sometimes cause diarrhea and vomiting when ingested in folk [14].

Schaeff [15] originally placed the type species in *Agaricus*; some researchers hold a different idea that the species is transferred into *Cortinellus*, *Gymnopus*, *Gyrophila*, *Pleurotus*, and *Tricholoma*, in different stages [16,17]. It was assigned by Singer to *Tricholomopsis* until 1939 and selected *T. rutilans* as the type species to accommodate the saprophytic tricholomatoid group [18,19]. Meanwhile, *Agaricus sulphureoides* Peck and *A. flavescens* Peck were transferred to *Tricholomopsis*, as *T. sulphureoides* Singer, and *T. flavescens* (Peck) Singer [20]. However, ‘*T. rutilans*’ may be a species complex and should be reclassified based on molecular and morphological characteristics by recent studies [7,9,21]. A total of 88 records are shown in the Index Fungorum (https://indexfungorum.org/Names/Names.asp, accessed on 6 November 2024). However, only about 60 species were accepted in the genus of *Tricholomopsis* including five variants, *T. bambusina* var. *bambusina*, *T. elegans* var. *elegans*, *T. ornata* var. *ornata*, *T. rutilans* var. *rutilans*, *T. sulfureoides* var. *sulfureoides*, all over the world. These species are found in various climates from the boreal to tropical regions, primarily in the Northern Hemisphere [22]. Approximately 20 species of ITS and nLSU genes were submitted to the Genbank database [9]. Furthermore, there are 11 species lacking molecular data that require additional collection of type materials and systematic research [23]. The primary method for species identification in *Tricholomopsis* is through phylogenetic analysis of ITS and nLSU data. In addition, with the application of molecular techniques, the sequences of many fossil specimens were used to speculate on the divergence time of the species. The molecular clock (MC) hypothesis states that the evolutionary rate of DNA or protein sequences is approximately constant among species, and the divergence time of species can be obtained by comparison with fossil information [24]. Furthermore, molecular phylogeny and divergence time estimates for major rodent groups have been also applied to higher-order taxonomic systems [25]. Divergence times of fungi advances have emerged as a crucial criterion for determining difficult-to-classify genera within the fungal taxonomy. Recently, the outline of Basidiomycota was established based on the phylogenomic relationships and divergence times of main higher taxonomic units [26,27]. Hu et al. [28] carried out the first comprehensive divergence-times estimation of the genera in Omphalotaceae and redefined *Gymnopus* s.l. and related genera. Global patterns of mushroom evolution have been generated by Varga et al. [29], focusing on the fungal formation of Agaricomycetes and exploring the historical events of species differentiation and extinction. The application of differentiation time in fungal taxonomic studies is becoming increasingly common, with several findings regarding the divergence timing, origin, and dispersal of macrofungi [30,31,32]. However, the divergence time of *Tricholomopsis* remains unclear.

Recently, while studying the diversity of macrofungi in Gansu Province in 2022, five tricholomatoid-like specimens were found in northwestern China, which were identified as an undescribed taxon and were temporarily placed in the family Phyllotopsidaceae, based on their physical characteristics and the ITS and nLSU dataset combined with divergence time, according to the current the phylogenetic framework of Phyllotopsidaceae.

## 2. Materials and Methods

### 2.1. Morphological Studies

The specimens were collected from Gansu Province in northwestern China. Macro-morphological and microscopic characteristics were described following [9,19]. Fresh basidiomata were photographed by Sony ZV-E10L in the field and macroscopic characters were noted. The special color terms followed Petersen [33]. The dried specimens were deposited in the Fungal Herbarium of Gansu Agricultural University (MHGAU, Lanzhou, China). Microscopic characters were examined from dried material by mounting hand-cut sections of the basidiomata and photoed in 5% KOH for 2 min, then stained with 2% Phloxine B (C_20_H_4_Br_4_Cl_2_K_2_O_5_). In addition, the sections were prepared in Melzer’s reagent (IKI) and Cotton Blue (CB). All measurements were performed in 2% Phloxine B. Sections were studied at a magnification × 1000 using Olympus Corporation BX63 (Beijing, China) and photographed using Olympus Corporation DP75. Then 5% of measurements were excluded from each end of the range and given in parentheses when presenting spore size variation. The abbreviations were used in the text as follows: IKI = Melzer’s reagent, IKI+ = amyloid, IKI− = neither amyloid nor dextrinoid, KOH = 5% potassium hydroxide, CB = Cotton Blue, CB+ = cyanophilous, CB− = acyanophilous, L = mean length (arithmetic average of all basidiospores length), W = mean width (arithmetic average of all basidiospores width), Q = L/W ratio for each specimen studied, *n* (a/b) = number of spores (a) measured from given number of specimens (b).

### 2.2. DNA Extraction, PCR Amplification, and Sequencing

A CTAB rapid plant genome extraction kit (Aidlab Biotechnologies, Beijing, China) was used to obtain genomic DNA from dried specimens after pretreatment using TissuePrep (Jie Ling, TianJin, China), according to the manufacturer’s instructions with some modifications [34]. The internal transcribed spacer regions (ITS), 28S nuclear large subunit rDNA (nLSU) sequences were amplified with primer pairs ITS5/ITS4 [35] and LR0R/LR7 [36]. The PCR cycling schedule for ITS included an initial denaturation at 95 °C for 3 min, followed by 34 cycles at 94 °C for 40 s, 54 °C for 45 s, 72 °C for 1 min, and a final extension of 72 °C for 10 min, 4 °C forever; for nLSU, initial denaturation at 94 °C for 1 min, followed by 34 cycles at 94 °C for 30 s, 50 °C for 1 min, 72 °C for 1.5 min, and a final extension of 72 °C for 10 min, 4 °C forever [34]. The PCR products were purified and sequenced in Beijing Tsingke Biotech Co., Ltd., China (Beijing, China) with the same primers.

### 2.3. Phylogenetic Analyses

Newly generated sequences and related sequences downloaded from GenBank (Table 1) in this study using MAFFT 7.0 online service with the Q-INS-i strategy (http://mafft.cbrc.jp/alignment/server/, accessed on 11 September 2024) [37] under the default parameters and manually adjusted in BioEdit [38]. Positions deemed ambiguous to align were excluded manually to optimize the alignment. The aligned ITS and nLSU sequences were concatenated and transformed into the format using Mesquite version 3.2 [39]. The ITS + nLSU dataset was employed for the phylogenetic analysis and *Pluteus romellii* (Britzelm.) Sacc. was selected as the outgroup [19]. The final concatenated sequence alignments were submitted in TreeBase (https://treebase.org/treebase-web/home.html, accessed on 8 September 2024; submission ID 30798) for the ITS + nLSU dataset and the taxonomic novelties in MycoBank (http://www.MycoBank.org, accessed on 8 September 2024).

Phylogenetic constructions of Maximum likelihood (ML), Maximum parsimony (MP), and Bayesian analyses (BI) were performed in the IQ-Tree v2.2.2.6 [40], in PAUP* 4.0b10 [41] and Mrbayes in PhyloSuite v1.2.3 [42], respectively. The GTR + I + G model was estimated as an optimal substitution model by jModelTest 2.3 using the corrected Akaike information criterion (AIC) [43]. Max-trees were reset to 5000 branches, if the minimum branch length is zero (amb-), collapsed, and all parsimonious trees were saved. ML bootstrap replicates (1000) were calculated in IQ-Tree using a rapid bootstrap analysis and search for the best-scoring ML tree. MP trees were inferred using the heuristic search option with tree bisection reconnection (TBR) branch swapping and 1000 random sequence additions. All characters were equally weighted and gaps were treated as missing data. Clades robustness was assessed using a bootstrap (BT) analysis with 1000 replicates [44]. Four Markov chains were run for two runs from random starting trees for 2 million generations until the split deviation frequency value <0.01, and trees were sampled every 100 generations. The first quarter generations were discarded as burn-ins. A majority rule consensus tree of all remaining trees was calculated for BI [34]. Other parameters in the ML and BI analysis used default settings. Branches were considered as significantly supported if they received bootstrap support for Maximum parsimony (BP) greater than or equal to 50%, Maximum likelihood (BP) greater than or equal to 70%, and Bayesian posterior probabilities (BPP) greater than or equal to 0.90. Trees were visualized with FigTree v1.4.3cc [45].

### 2.4. Divergence Time Analyses

The divergence times of two new families were estimated with the BEAST v2.6.5 software package [46] based on ITS sequence representing all main lineages in Basidiomycota. jModelTest with calculation under Akaike information criterion was used to estimate the best-fit evolutionary model for each alignment subjected to phylogenetic analysis. Two-time points were selected for calibration: (1) 90 million years ago (Mya) representing the minimum age of Agaricales by Archaeomarasmius leggetti, a fossil agaricoid species preserved in a Dominican amber; (2) 125 Mya representing the minimum age of Hymenochaetaceae by *Quatsinoporites cranhamii*, a fossil poroid species collected from Apple Bay on Vancouver Island. A Yule speciation model was selected as prior assuming a constant speciation rate per lineage and the uncorrelated lognormal relaxed clock model was applied. According to these time points, the offset age with a gamma distribution prior (scale = 20, shape = 1) for Agaricales was set as 90 Mya and for Hymenochaetaceae as 125 Mya. According to these time points, the offset age with a gamma distribution prior (scale = 20, shape = 1) for *Agaricales* was set as 90 Mya, for *Hymenochaetaceae* as 125 Mya. After 50 million generations, the first 10% of the sampled trees every 1000th generation were removed as burn-in and a Maximum clade-credibility (MCC) tree was summarized using Treeannotator v2.6.2. The resulting log file was checked for chain convergence using Tracer 1.5.

## 3. Results

### 3.1. Molecular Phylogeny

The ITS + nLSU dataset included sequences from 65 fungal collections representing 29 species. The dataset had an aligned length of 2152 characters, of which 1316 characters are constant, 184 are variable and parsimony-uninformative, and 652 are parsimony-informative. MP analysis yielded two parsimonious trees (TL = 2354, CI = 0.555, RI = 0.822, RC = 0.456, HI = 0.445). The best model for the ITS + nLSU combined dataset estimated and applied in the ML and BI was GTR + I + G. For Bayesian analysis: lset nst = 6, rates = invgamma; Ngammacat = 4, prset statefreqpr = fixed(empirical). Bayesian analysis resulted in a similar topology that of MP and ML analysis, with 2 million generations and an average standard deviation of split frequencies = 0.000033, and thus only the ML tree was provided. Both MP (BP) values (≥50%), ML (BP) values (≥70%), and BI (BPPs) (≥0.90) are shown at the nodes (Figure 1). The phylogeny results show that the newly collected specimens form a lineage sister with *Conoloma* and formed a distinct and independent lineage in the phylogeny analysis based on ITS + nLSU.

### 3.2. The Divergence Time of Neotricholomopsis

In the phylogenetic analyses of Phyllotopsidaceae and related taxa, the ITS dataset included 70 collections, of which 52 belonged to Agaricales. This dataset resulted in a concatenated alignment of 8330 characters with GTR + I + G as the best-fit evolutionary model. Chain convergence was indicated by the ESSs. In Agaricales, the genus *Neotricholomopsis* occurred with a mean crown age of 229.4 Mya with a 95% highest posterior density (HPD) of 129.63–343.08 Mya, followed by *Tricholomopsis*, which is most deeply diversified during the Paleogene, with a mean crown age of 163.76 Mya and a 95% HPD of 87.95–250.44 Mya (Figure 2).

### 3.3. Taxonomy

*Neotricholomopsis* B.Y. Wang and L.F. Fan, gen. nov.

MycoBank No. 856298

Etymology: referring to the tricholomatoid-like basidioma.

Diagnosis: compared to the genus *Tricholomopsis*, *Neotricholomopsis* has a fibrillose and glabrous stipe with a pinkish buff to salmon color without scales, but with a white fibrillose annuli-form zone on the upper part. The difference between *Neotricholomopsis* and *Conoloma* lies in the disc being slightly depressed and densely covered with peach to orange-red small scales, being fibrillose and glabrous with a cream to light pinkish yellow color [2]. *Neotricholomopsis* is characterized by being fibrillose and glabrous, with orange-red scales slightly depressed in the disc of the pileus, with a white fibrillose annuli-form zone on the upper part of the stipe in Phyllotopsidaceae.

Habitat: growing in high-altitude coniferous forests woodland at about 3000 m.

Description: Basidioma annual, tricholomoid, small to medium sized, fleshy, fibrillose and glabrous, with orange-red scales slightly depressed in the disc of the pileus, with a white fibrillose annuli-form zone on the upper part of the stipe. The basidia are narrowly clavate to subclavate, usually four spored. Basidiospores are small, globose to subglobose, colorless and hyaline, smooth, CB+, IKI−. Cheilocystidia are abundant. Pileipellis a cutis. Clamp connections are present.

Known distribution: only known from northwestern China.

***Neotricholomopsis globispora* B.Y. Wang and L.F. Fan, sp. nov.** Figure 3 and Figure 4.

MycoBank No. 856299

Etymology: referring to the shape of basidiospores as globose to subglobose.

Diagnosis: *N. globispora* differs from other species in *Tricholomopsis* by having a slightly depressed disc with peach to salmon color small scales, fibrillose and glabrous pileus with a cream to pinkish buff, lamellae adnate to sub-decurrent, stipe that are fibrillose and glabrous with a pinkish buff to salmon color and without scales, but with a white fibrillose annuli-form zone on the upper part. Basidiospores are mostly globose to subglobose, cheilocystidia obvious, narrowly clavate, growing in coniferous forests.

It resembles *T. flammula* Métrod ex Holec and *T. depressa* Zhu L. Yang and G. S. Wang in pileus. But the former can be differentiated from *T. globispora* by having ellipsoid, larger basidiospores (5.6–8.0 × 3.2–4.8 μm vs. 5.0–6.0 × 4.0–5.0 μm, Q = 1.51–1.63 vs. 1.11–1.13), and the latter differs from this new species by having ellipsoid to elongate, larger basidiospores (5.5–7.5 × 3.5–5.0 μm vs. 5.0–6.0 × 4.0–5.0 μm, Q = 1.38–2.00 vs. 1.11–1.13) and stipe with covered grey pink to pink scales or fibrils [2,47]. *N. globispora* differs from species in *C. mucronatum* by having fibrillose pileus and disc slightly depressed with small scales, without mucronate umbo [21].

Holotype: CHINA. Gansu Province: Tibetan Autonomous Prefecture of Ganan, Taohe National Nature Reserve, 34°40′70″ N, 103°53′26″ E, in coniferous forest land, 7 September 2022, FLF180.

Basidiomata annual, growing singly or in clusters. The pileus is 44–73 mm in diameter, convex or hemispherical when young, nearly hemispherical with a waved margin, the edges are more than lamellae, the disc is slightly depressed with peach to salmon-color small scales, and is fibrillose and glabrous with a cream to pinkish buff when at maturity. Context cream, up to 1 mm wide, without an obvious change when cut. Lamellae are up to 3 mm wide, adnate to sub-decurrent, a pinkish buff to salmon color, rather crowded, with four to seven relatively regular rows of lamellulae, and are thin. The stipe is 48–84 × 8–14 mm, central, cylindrical, hollow, frequently straight to curved, fibrillose and glabrous, with a color similar to the lamellae when fresh and with a white fibrillose annuli-form zone on the upper part, annulus, and volva absent.

Basidiospores are mostly globose to subglobose, smooth, hyaline, thin-walled, IKI−, CB+, 5.0–6.0 × (3.5–) 4.0–5.0 (–5.5) μm, L = 5.1 μm, W = 4.5 μm, Q = 1.11–1.13 (*n* = 60/2). Basidia are 19.0–38.0 × 4.0–7.0 μm, narrowly clavate to subclavate, terminal inflated sometimes, sterigmata up to 5 μm long, smooth, hyaline, thin-walled, with a clamp connection at the base. Cheilocystidia are obviously prominent, mostly clavate to broadly clavate, smooth, hyaline, 34.0–60.0 × 6.0–9.0 μm. Pleurocystidia are absent. Pileipellis a cutis, composed of hyphae 4.0–8.5 μm in diameter, hyaline, slightly inflated in the KOH; Stipitipelis a cutis, hyphae 1.5–6.0 μm in diameter, hyaline, caulocystidia are absent. Clamp connections are frequently observed in different tissues.

Additional specimens examined (paratype): CHINA. Gansu Province: Tibetan Autonomous Prefecture of Ganan, Taohe National Nature Reserve, 34°40′70″ N, 103°53′26″ E, in coniferous forest land, 17 August 2022, FLF286, 16 September 2023, FLF918, FLF923, FLF936.

## 4. Discussion

*Neotricholomopsis globispora* has been identified through a comprehensive analysis of both morphological and molecular evidence, which is characterized by its nearly hemispherical basidiomata with waved margin, disc slightly depressed, and densely covered with peach to orange-red small scales, being fibrillose and glabrous with a cream to light pinkish yellow color, basidiospores that are mostly globose to subglobose, 5.0–6.0 × (3.5−) 4.0–5.0 (−5.5) μm, Q = 1.11–1.13, IKI−, CB+, cheilocystidia are obvious, mostly narrowly clavate, 34.0–60.0 × 6.0–9.0 μm, pleurocystidia and caulocystidia absent, and growing in the high-altitude coniferous forests.

Macro-morphologically, *N. globispora* resembles *T. flammula* by having densely covered small scales, and adnate to sub-decurrent lamellae. However, *T. flammula* has small-sized basidiomata with greenish-yellow, yellow to ochre-yellow convex to depressed pileus, densely covered with violet-red, purple-red to red-brown small scales, pale greenish yellow, yellow to ochre-yellow lamellae, and ellipsoid, less frequently obovoid-ellipsoid, ovoid-ellipsoid to oblong, larger basidiospores (5.6–8.0 × 3.2–4.8 μm vs. 5.0–6.0 × 4.0–5.0 μm, Q = 1.51–1.63 vs. 1.11–1.13) [47]. *N. globispora* resembles *T. depressa* by having medium-sized basidiomata, often depressed at the center, margin involute, densely covered scales, and lamellae that are adnate to sub-decurrent. However, *T. depressa* has margin of the pileus and stipe is also densely covered with grey-pink to pink small scales, yellow-white to yellowish lamellae, and obviously larger basidiospores (5.5–7.5 × 3.5–5.0 μm vs. 5.0–6.0 × 4.0–5.0 μm, Q = 1.38–2.00 vs. 1.11–1.13) [2].

Microscopically, *T. rutilans* is very similar to *N. globispora* by having subglobose to ellipsoid basidiospores, presenting obvious cheilocystidia and absence of caulocystidia. However, the former differs from *N. globispora* in that *T. rutilans* mainly distributed in Europe, the pileus and stipe are densely covered with red-violet fibrils to scales, and pleurocystidia are absent or rare [48]. In addition, they formed two independent lineages in the phylogenetic analysis (Figure 1).

In addition, *Conoloma mucronata* Zhu L. Yang and G.S. Wang and *Tricholomopsis yunnanensis* (M. Zang) Li R. Liu, Yan C. Li and Zhu L. Yang were initially described from Yunnan province, and also grow in woodlands of coniferae. However, *C. mucronate* can be differentiated from *N. globispora* by its plano-cobvex glabrous pileus, mucronate or coracoid dark ochraceous yellow umbo, lamellae sinuate to decurrent, and significantly larger basidiospores (5.5–6.5 × 5.0–6.0 vs. 5.0–6.0 × 4.0–5.0 μm, Q = 1.04–1.20 vs. 1.11–1.13). Moreover, *N. globispora* is absent of caulocystidia while *C. mucronata* has abundant caespitose caulocystidia [21]. *T. yunnanensis* is easily differentiated from *N. globispora* by its hemispherical pileus and cylindrical stipe covered with red-violet to red-brown fibrils, shorter and slightly broader basidia (22.0–29.0 × 6.0–8.0 vs. 19.0–38.0 × 4.0–7.0 μm), and most important is that *T. yunnanensis* has subfusiform or cylindrical to narrowly clavate pleurocystidia [11,45]. Pleurocystidia is undetected in our materials.

The ITS + nLSU phylogenetic analysis (Figure 1) showed that these newly collected specimens formed a considerably supported lineage and were independent of other species in *Tricholomopsis* and *Conoloma* (94/95/1), which means there are more species in the *Neotricholomopsis* to be discovered, especially in this group. Consequently, combined morphology, molecular data, and eco-geographical distribution are obligated to distinguish species, and ITS + nLSU datasets are normally selected for species delimitation in this genus. However, the dispute over the higher taxonomy status of *Neotricholomopsis* needs more and more different regional materials and more loci to conduct a comprehensive analysis to clarify its phylogenetic relationship with different genera.

Currently, origin time analyses have offered valuable insights into the evolution of macrofungi. The time ranges for basidiomycota, with the phylum originating ca. 530 Mya, the subphyla 406–490 Mya, most classes 245–393 Mya and orders 120–290 Mya were inferred by He et al. [26]. In our study, divergence time is estimated with ITS sequences representing all main lineages within Agaricales (Figure 2). The MCC tree shows that Basidiomycota occurs in a mean stem age of 419.4 Mya, according to previous research [26,31,49]. The time range of Agaricales was 90–247 Mya. Our study suggests that the new genus *Neotricholomopsis* possibly emerged at 229.36 Mya latter Triassic (PP = 1). Considering the divergence estimation of most genera in Agaricales is at 2–182 Mya [27] earlier than the occurrence date of *Neotricholomopsis*, this estimation of our new genus seems reasonable. In the late Triassic, extraterrestrial impact, along with the resulting fluctuations in temperatures and the migrate of coniferous trees [50], may have contributed to the differentiation of *Tricholomopsis* and *Neotricholomopsis*, just like many macrofungi [51,52]. At the species level, divergence of most *Tricholomopsis* and *Neotricholomopsis* species occurred during the Eocene and Oligocene. This divergence is speculated to be associated with the fluctuating temperatures and the increasing emergence of angiosperms and the migration of tree of coniferous trees during that period [27,53].

## Figures and Tables

**Figure 1 jof-10-00784-f001:**
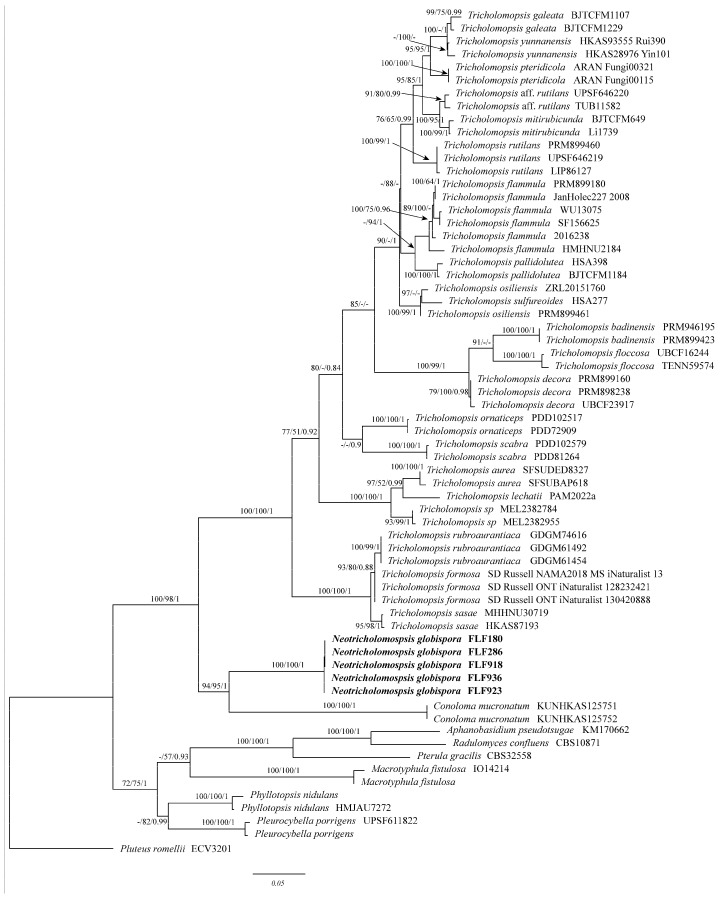
This is a MrBayes tree that shows the phylogenetic relationship of Phyllotopsidaceae based on the combined ITS + nLSU dataset. The bootstrap support values for ML, MP, and Bayes are given at each node, with values greater than 70% for ML and 50% for MP, and greater than 0.9 for Bayes. The new species are shown in bold.

**Figure 2 jof-10-00784-f002:**
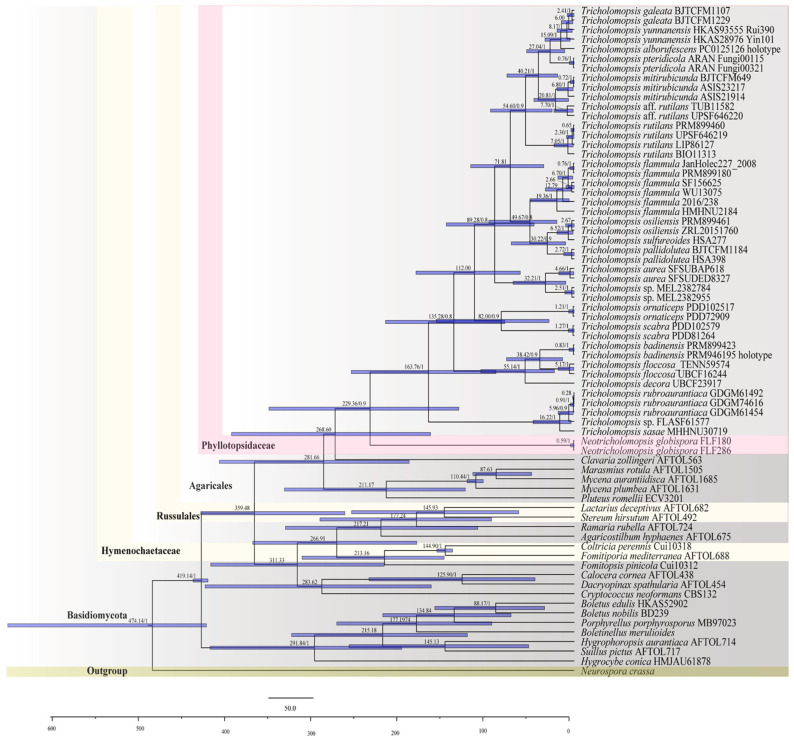
Chronogram and estimated divergence times of *Neotricholomopsis* generated by molecular clock analysis using the ITS dataset. The chronogram was obtained using the Basidiomycota divergence time of 400 Mya as the calibration point. The calibration points and objects of this study are marked in the chronogram. The geological time scale is millions of years ago (Mya).

**Figure 3 jof-10-00784-f003:**
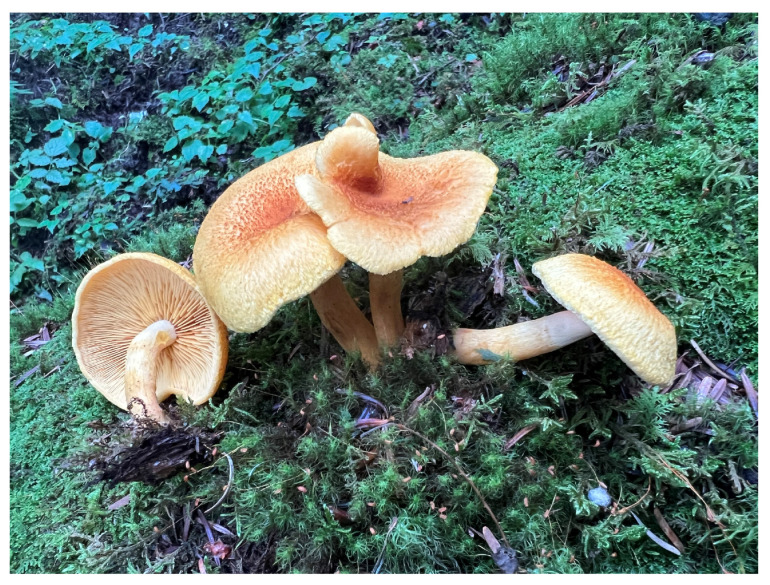
Basidiomata of *Neotricholomopsis globispora* (FLF180, holotype).

**Figure 4 jof-10-00784-f004:**
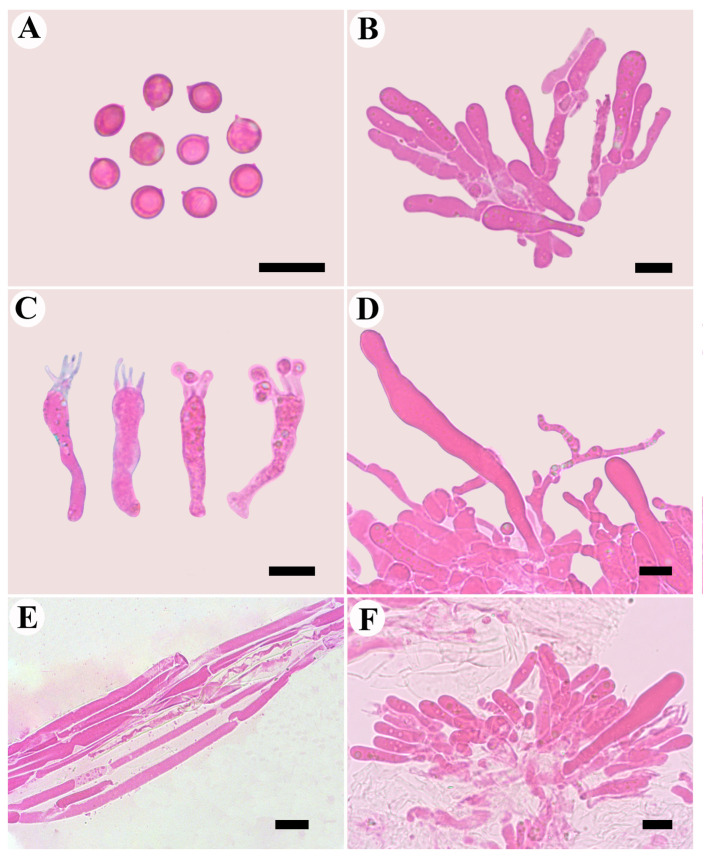
Microscopic structures of *Neotricholomopsis globispora* (FLF180, holotype). (**A**) basidiospores, (**B**) probasidia, (**C**) basidia, (**D**) cheilocystidia, (**E**) hypha with clamp connections, (**F**) hymenium; Scale bars = 10 μm.

**Table 1 jof-10-00784-t001:** Specimens used in molecular phylogenetic studies and their GenBank accession numbers.

Species Name	Voucher Number	Country of Origin	Ganbank Accession No	
			ITS	LSU
*Aphanobasidium pseudotsugae*	K(M) 170662	United Kingdom	MK953243	MK953402
*Conoloma mucronatum*	Yang 6827	China	OP627093	OP604167
*C. mucronatum*	GLGLJ173	China	OP627092	OP604166
*Macrotyphula fistulosa*	IO. 14.214	Spain	MT232352	KY224088
*M. fistulosa*	MA-Fungi 47944	Spain	AJ296348	AY463441
** *Neotricholomopsis globispora* **	**FLF180**	**China**	**OR651765**	**OR607663**
** *N. globispora* **	**FLF286**	**China**	**OR651766**	**OR607664**
** *N. globispora* **	**FLF918**	**China**	**PQ130423**	**PP907041**
** *N. globispora* **	**FLF923**	**China**	**PQ130424**	**PP907042**
** *N. globispora* **	**FLF936**	**China**	**PQ130425**	**PP907043**
*Phyllotopsis nidulans*	IO. 14.214	Spain	MT232308	DQ071736
*Ph. nidulans*	HMJAU 7272	China	GQ142019	GQ142039
*Pleurocybella porrigens*	UPSF 611822	Spain	MT232355	MT232309
*Pl. porrigens*	UBCF 33075	Canada	MF908476	DQ071737
*Pluteus romellii*	AFTOL-ID625	USA	AY854065	AY634279
*Pterula gracilis*	CBS: 325.58	Netherlands	MH857800	MH869334
*Radulomyces confluens*	CBS: 108.71	Germany	MH860025	MH871809
*Tricholomopsis aff. rutilans*	UPSF 646220	Sweden	KP058984	KP058985
*T. aff. rutilans*	TUB 11582	Sweden	KP058981	
*T. aurea*	SFSU: DED8327	USA	MF100960	MF100993
*T. aurea*	SFSU: BAP618	USA	MF100961	MF100994
*T. badinensis*	PRM: 946195	Czech Republic	LS992163	
*T. badinensis*	PRM: 899423	Czech Republic	LS992164	
*T. decora*	UBCF 23917	Canada	KJ146732	
*T. decora*	PRM: 899160	Solvakia	HE649942	
*T. decora*	PRM: 898238	Czech Republic	FN554891	
*T. flammula*	PRM: 899180	Czech Republic	HE649940	
*T. flammula*	WU: 13075	Czech Republic	HE652866	
*T. flammula*	JanHolec: 227/2008	Czech Republic	FN554894	
*T. flammula*	SF 156625	Sweden	KP058975	KP058976
*T. flammula*	2016-238	China	KY820050	
*T. flammula*	HMHNU 2184	China	MF967218	
*T. floccosa*	TENN 59574	USA	AY329597	
*T. floccosa*	UBC F16244	Canada	EF530924	
*T. formosa*	IN 128232421	USA	OP643156	
*T. formosa*	IN 13892104	USA	OP541626	
*T. formosa*	IN 130420888	USA	OP643359	
*T. galeata*	BJTCFM 1107	China	MW871732	MW871622
*T. galeata*	BJTCFM 1229	China	MW871782	MW871630
*T. lechatii*	LIP0202264	France	OM793061	OM793062
*T. mitirubicunda*	BJTCFM 649	China	MW871620	MW871823
*T. mitirubicunda*	HKAS 101060	China	ON641527	ON627759
*T. ornaticeps*	PDD: 102517	New Zealand	KY010822	
*T. ornaticeps*	PDD: 72909	New Zealand	KY010817	KY010826
*T. osiliensis*	PRM: 899461	Czech Republic	HE649943	
*T. osiliensis*	ZRL 20151760	China	LT716068	KY418884
*T. pallidolutea*	HAS 398	China	MW871614	MW871640
*T. pallidolutea*	BJTCFM 1184	China	MW871749	MW871631
*T. pteridicola*	ARAN-Fungi 00321	Sweden	KP058992	KP058993
*T. pteridicola*	ARAN-Fungi 00115	Sweden	KP058994	KP090343
*T. rubroaurantiaca*	GDGM 74616	China	MN912496	MN912493
*T. rubroaurantiaca*	GDGM 61454	China	MN912494	MN912491
*T. rubroaurantiaca*	GDGM 61492	China	MN912495	MN912492
*T. rutilans*	PRM: 899460	Czech Republic	HE649946	
*T. rutilans*	UPSF 646219	USA	NR_172749	NG_071241
*T. rutilans*	LIP86127	Sweden	KP058995	
*T. sasae*	MHHNU 30719	China	MK388148	
*T. sasae*	HKAS 87193	China	ON641576	
*T. scabra*	PDD: 102579	New Zealand	KY010823	KY010832
*T. scabra*	PDD: 81264	New Zealand	KY010819	KY010830
*Tricholomopsis* sp.	MEL: 2382784	Australia	KP012965	
*Tricholomopsis* sp.	MEL: 2382955	Australia	KP012816	
*T. sulfureoides*	HSA 277	China	MW867238	MW871623
*T. yunnanensis*	HKAS 28976, Yin101	China	MZ470249	
*T. yunnanensis*	HKAS 93555, Rui390	China	MZ470248	OL614951

Note: Newly generated sequences are in bold.

## Data Availability

The original contributions presented in the study are included in the article, further inquiries can be directed to the corresponding authors.

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
