# Peer review of "A New Genus Neotricholomopsis Gen. Nov and Description of Neotricholomopsis globispora Sp. Nov. (Phyllotopsidaceae, Agaricales) from Northwestern China Based on Phylogeny, Morphology, and Divergence Time"

_jof, 2024, doi:10.3390/jof10110784_

Round 1

Reviewer 1 Report

The phylogenetic relationships among some suborders of the order Agaricales are largely unresolved, and the phylogenetic positions and delimitations of some taxa including Tricholomopsis remain unsettled. Until 2023, research has not provided firm evidence for the phylogenetic position of Tricholomopsis. Very recent works reported seventeen Tricholomopsis species in China, including eight novel species and one novel variety, by the year 2023. Such species diversity is underestimated because of morphological similarity of the different taxa. The reviewed manuscript describes Neotricholomopsis gen. nov, and N. globispora sp. nov.

1. The substrate preference of Tricholomopsis species and the transitions of the pileate ornamentations among the species were discussed in recent research. The authors wrote (Lines 45-46) that most species are usually associated with decaying wood, inhabit conifer wood, hardwood, or bamboo. What are for the substrate preference of Neotricholomopsis species within the genus, please discuss if possible.

2. There are few morphological attributes in the discussed species (Neotricholomopsis globispora) that are similar to those in other fungi. Neotricholomopsis globispora resembles Tricholomopsis flammula and Tricholomopsis depressa in pileus (Lines 211, 258, 264). Besides, Neotricholomopsis globispora is very similar to Tricholomopsis  rutilans (microscopically; Line 270).

However, while T. flammula and T. rutilans are evidenced, on the basis of literary data, to form lineages in the phylogenetic analysis independent of that for Neotricholomopsis globispora (Figure 1), the species Tricholomopsis depressa is absent in the phylogenetic relationship of Phyllotopsidaceae based on the combined ITS+nLSU dataset. Do the authors agree that it is a serious drawback of their manuscript content? Please consider a possibility of providing more extended reasonings.

Author Response

Dear Editor and Reviewers,

Thank you for offering us an opportunity to improve the quality of our submitted manuscript “A new genus Neotricholomopsis gen. nov and description of Neotricholomopsis globispora sp. nov. (Phyllotopsidaceae, Agaricales) from northwestern China based on phylogeny, morphology, and divergence time”. We appreciated very much the reviewer’s constructive and insightful comments. In this revision, we have address all of these comments. We hope the revised manuscript has now met the publication standard of your journal.

We highlighted all the revisions in yellow color.

On the next pages, our point-to-point responses raised by the reviewer are listed.

Review1:

Comment 1: The substrate preference of Tricholomopsis species and the transitions of the pileate ornamentations among the species were discussed in recent research. The authors wrote (Lines 45-46) that most species are usually associated with decaying wood, inhabit conifer wood, hardwood, or bamboo. What are for the substrate preference of Neotricholomopsis species within the genus, please discuss if possible.

Response: It is added in line 46.

Comment 2: There are few morphological attributes in the discussed species (Neotricholomopsis globispora) that are similar to those in other fungi. Neotricholomopsis globispora resembles Tricholomopsis flammula and Tricholomopsis depressa in pileus (Lines 211, 258, 264). Besides, Neotricholomopsis globispora is very similar to Tricholomopsis rutilans (microscopically; Line 270). However, while T. flammula and T. rutilans are evidenced, on the basis of literary data, to form lineages in the phylogenetic analysis independent of that for Neotricholomopsis globispora (Figure 1), the species Tricholomopsis depressa is absent in the phylogenetic relationship of Phyllotopsidaceae based on the combined ITS+nLSU dataset. Do the authors agree that it is a serious drawback of their manuscript content? Please consider a possibility of providing more extended reasonings.

Response: Tricholomopsis depressa is absent in the phylogenetic relationship due to it being a new species described by Zhu L. Yang & G.S. Wang 2023 based on single-copy orthologous genes (Only the 10 records of TEF 1 gene is available in the NCBI). So we discussed them based on their morphological characteristics. And what’s more, Tricholomopsis depressa clustered in Sect. Tricholomopsis the phylogenetic relationship of Phyllotopsidaceae (Wang et al. 2023). In our phylogeny, Sect. Tricholomopsis and Neotricholomopsis formed two independent lineages.

Reviewer 2 Report

Abstract and Introduction.

The abstract does not make clear the importance of divergence time and its connection to fungal taxonomy. Furthermore, from a broader perspective, the scientific contribution of this study is not clear.

Please place the taxonomic novelty of this study in the context of how it helps us understand evolutionary relationships within the Phyllotopsidaceae.

Materials and Methods (Lines 139-151): Overall clear, but details on divergence time analysis are somewhat sparse. Need to elaborate a bit more on the rationale for calibration points and how divergence time estimates fit into the broader context of Agaricales evolution.

Results and Discussion (Lines 173-300):

The phylogenetic results are well presented, but not well discussed in significance. The implications of divergence time analysis for the new species and genus are not clearly defined. - Emphasize why Neotricholomopsis is distinct at the genus level vs. Tricholomopsis; a more detailed discussion of divergence time results is needed.

The discussion repeats information and does not adequately compare the new species to closely related species. It mentions phylogenetic placement and divergence times, but poorly explores their significance in a taxonomic context.

Author Contributions are missing (line 302).

Mycobank Registration Number is missing, usually this registration is provided provisionally before publication. (line 188, 205).

Author Response

Response letter

Dear Editor and Reviewers,

Thank you for offering us an opportunity to improve the quality of our submitted manuscript “A new genus Neotricholomopsis gen. nov and description of Neotricholomopsis globispora sp. nov. (Phyllotopsidaceae, Agaricales) from northwestern China based on phylogeny, morphology, and divergence time”. We appreciated very much the reviewer’s constructive and insightful comments. In this revision, we have address all of these comments. We hope the revised manuscript has now met the publication standard of your journal.

We highlighted all the revisions in yellow color.

On the next pages, our point-to-point responses raised by the reviewer are listed.

Review2:

Abstract and Introduction.

Comment 1: The abstract does not make clear the importance of divergence time and its connection to fungal taxonomy. Furthermore, from a broader perspective, the scientific contribution of this study is not clear.

Response 1: We have revised the introduction as “Furthermore, molecular phylogeny and divergence time estimates for major rodent groups have been also applied to higher-order taxonomic systems (Adkins et al. 2001). divergence times of fungi advances have emerged as a crucial criterion for determining difficult-to-classify genera within the fungal taxonomy. Recently, the outline of Basidiomycota was established based on the phylogenomic relationships and divergence times of main higher taxonomic units (He et al. 2019; He et al. 2024). Hu et al. (2024) do the first comprehensive divergence-times estimation of the genera in Omphalotaceae and redefined Gymnopus s.l. and related genera. Global patterns of mushroom evolution have been generated by Varga et al. (2019), focusing on the fungal formation of Agaricomycetes and exploring the historical events of species differentiation and extinction. The application of differentiation time in fungal taxonomic studies is becoming increasingly common, with several findings regarding the divergence timing, origin and dispersal of macro-fungi. However, the divergence time of Tricholomopsis remains unclear.

Please place the taxonomic novelty of this study in the context of how it helps us understand evolutionary relationships within the Phyllotopsidaceae.

Comment 2: Materials and Methods (Lines 139-151): Overall clear, but details on divergence time analysis are somewhat sparse. Need to elaborate a bit more on the rationale for calibration points and how divergence time estimates fit into the broader context of Agaricales evolution.

Response 2: We have added the relevant operating parameters in the Materials and Methods.

Results and Discussion (Lines 173-300):

Comment 3: The phylogenetic results are well presented, but not well discussed in significance. The implications of divergence time analysis for the new species and genus are not clearly defined. - Emphasize why Neotricholomopsis is distinct at the genus level vs. Tricholomopsis; a more detailed discussion of divergence time results is needed.

 Response 3: We have added the divergence time analysis for the new genus and add the discussion about the distinction of Neotricholomopsis and Tricholomopsis.

“Our study suggests that the new genus Neotricholomopsis possibly emerged at 229.36 Mya latter Triassic (PP = 1). Considering the divergence estimation of most genera in Agaricales is at 2-182 Mya (He et al. 2024) earlier than the occurrence date of Neotricholomopsis, this estimation of our new genus seems reasonable. In the late Triassic, extraterrestrial impact, along with the resulting fluctuations in temperatures and the migrate of coniferous trees (O’Keefe & Ahrens 1989), may have contributed to the differentiation of Tricholomopsis and Neotricholomopsis, just like many macrofungi (Zamora & Ekman 2020; Zhao et al. 2023).”

Comment 4: The discussion repeats information and does not adequately compare the new species to closely related species. It mentions phylogenetic placement and divergence times, but poorly explores their significance in a taxonomic context.

 Response 4: We have added the contents of the relationship of species and the divergence time in a taxonomic context.

“At the species level, divergence of most Tricholomopsis and Neotricholomopsis species occurred during the Eocene and Oligocene. This divergence is speculated to be associated with the fluctuating temperatures and the increasing emergence of angiosperms and the migration of tree of coniferous trees during that period (He et al. 2024, Zhou et al. 2024).”

Round 2

Reviewer 2 Report

The text has been improved in several areas, particularly in the discussion of divergence time analysis. However, I believe the description of the genus requires a stronger foundation. The description of phenotypic characters (morphology) remains deficient, and the description in the text and the figures need to be improved. Including metabolite data would also be beneficial.

A phylogeny based solely on rDNA (ITS + LSU) is quite limited, as ITS and LSU are linked loci.

The text has been improved in several areas, particularly in the discussion of divergence time analysis. However, I believe the description of the genus requires a stronger foundation. The description of phenotypic characters (morphology) remains deficient, and the description in the text and the figures need to be improved. Including metabolite data would also be beneficial.

A phylogeny based solely on rDNA (ITS + LSU) is quite limited, as ITS and LSU are linked loci.

Author Response

Review2:

The text has been improved in several areas, particularly in the discussion of divergence time analysis.

Comment 1: However, I believe the description of the genus requires a stronger foundation. The description of phenotypic characters (morphology) remains deficient, and the description in the text and the figures need to be improved. Including metabolite data would also be beneficial.

Response 1: Yes, it was added in lines 204-210212-214,224-226 respectively and all revisions are shown highlight with yellow.

Comment 2: A phylogeny based solely on rDNA (ITS + LSU) is quite limited, as ITS and LSU are linked loci.

Response 2: Yes, ITS and LSU are linked loci and they were widely used in phylogenetic relationship. ITS + LSU dataset is enough to illustrate the relationship in Phyllotopsidaceae (王庚申和杨祝良 2023Hosen et al. 2020; Holec et al. 2019 etc.). Our result of phylogeny is consistent with the results of previous studies such as Wang et al. (2023) which used single-copy orthologous. Furthermore, a part reported species in Phyllotopsidaceae no other genes were available from NCBI or no release such as TEF1-α, RPB1, or RPB2.

Thank you for the reviewer’s constructive and insightful comments again.